# How Childhood Socioeconomic Status Impacts Adult Food Preference: The Mediating Role of Stress and Trait Appetite

**DOI:** 10.3390/bs12070202

**Published:** 2022-06-22

**Authors:** Jim B. Swaffield, Qi Guo

**Affiliations:** 1School of Business, University of Alberta, Edmonton, AB T6G 2R6, Canada; 2Centre for Research in Applied Measurement and Evaluation, Department of Educational Psychology, University of Alberta, Edmonton, AB T6G 2G5, Canada; qig@ualberta.ca

**Keywords:** stress, eating, food, obesity, appetite, trait appetite, socioeconomic status, childhood, environmental harshness

## Abstract

Prior research has shown that adults who were raised in a low socioeconomic status (SES) environment are more likely to desire energy-dense foods. Research has also shown a positive correlation between current stress levels and the desire for energy-dense foods. We hypothesized that stress and *trait appetite* mediate the relationship between childhood SES and the desire for low and high-energy-dense foods. In this study, 311 adults participated in an online experiment in which they were shown images of five food items from each of the six major food categories (vegetables, fruits, grains, dairy, meat/poultry, and sweets) and rated how desirable each food item is. Next, we asked a series of questions that identified the participant’s sex, early childhood socioeconomic conditions, and current stress level. We also identified whether the participants had a trait or state eating personality. A path analysis was used to confirm the hypothesis that stress plays a mediating role between SES and food preference, and that an orderly relationship exists between these variables. The results show the hypothesis was supported and that the results were statistically significant. Specifically, the results show that the desire for low and high-energy-dense foods is indirectly influenced by one’s early childhood environment, and that food desirability is mediated by both stress and trait appetite. In addition, this analysis showed that in some situations, stress can both increase and decrease the desire for high-energy-dense foods. These findings also contribute to our understanding of how environmental conditions (safe and harsh environments) affect appetite and the desire for low and high-energy-dense foods. It also provided a deeper understanding of how these food choices can be adaptive under different ecological conditions.

## 1. Introduction

Excessive body weight and obesity are associated with a number of medical conditions such as metabolic disorders, heart disease, diabetes, cancer, stroke, and orthopedic problems [1]. In addition, carrying excessive body weight lowers life expectancy [2] and creates an economic burden as it increases the costs of delivering public healthcare services [3].

The etiology of obesity is complex as it is an outcome of an interaction between multiple genetic and environmental factors [4,5]. To gain a deeper understanding of the obesity epidemic, we need to understand not only what environmental factors are correlated with obesity, but also how environmental factors interact with each other, and how environmental factors interact with genetic factors. For example, it is well-established that early childhood socioeconomic status (SES) [6,7] and stress [8] are strongly correlated with obesity. However, what is less well-understood is how these factors interact and lead to specific food preferences and the volume of food desired.

With regards to obesity and psychosocial factors, there is a gap in the extant literature in terms of identifying what variables mediate the relationship between low childhood socioeconomic conditions, food preference, and the desire to eat. Understanding these interactions may help to develop strategies that promote healthier food consumption behaviour and reduce health problems associated with having excess body weight.

Therefore, in this paper, we examine the relationships between childhood and adult socioeconomic status (SES), stress, and trait appetite, and how these factors collectively influence food preference.

### 1.1. The Effects of Socioeconomic Status on Eating

Research has shown that harsh SES conditions experienced during childhood can adversely affect an individual’s physical and psychological development and create patterns of behaviour that are carried into adulthood. More specifically, early childhood SES conditions can influence how an adult responds to stress, their food preferences, the volume of food consumed [9,10,11], the likelihood that one will desire to eat in the absence of an energy deficit [12], the development of eating disorders [13], and the likelihood that one will suffer from adult obesity [6,7].

Studies have also shown that stress influences food preference and the volume of food consumed. Stress can both increase and decrease appetite depending on the type and intensity of the stressor [6,14]. Low-intensity chronic stress tends to increase appetite whereas acute stress tends to decrease appetite [15,16]. Mild stress tends to increase the desire to consume high-energy-dense food [16,17,18,19] and decrease the desire for low-energy-dense foods [20].

Socioeconomic status and stress both affect appetite and food choices. Studies that have looked at the association between SES and stress have produced mixed and sometimes contradictory results. For example, Macleod, Smith, Metcalfe, and Hart [21] concluded there is a robust correlation between income, education, functional health, and SES; however, they did not find a relationship between reported stress levels and SES. In contrast, research by Matthews, Gallo, and Taylor [22] and Macleod et al. [21] have found a strong association between low SES, stress, and poor health.

These inconsistent findings may be a result of two factors. First, research has shown that it is not simply having low SES that creates stress, but rather the subjective feeling of being inferior to others who are perceived to possess higher SES [23]. Sapolsky [24] adds a low SES environment does not necessarily lead to elevated stress. Rather, low SES generates elevated levels of stress when in the presence of others who have higher SES. Therefore, to understand the relationship between SES and stress, it must be evaluated relative to and in the presence of others who have the same or different levels of SES.

A second factor that may contribute to these inconsistent findings may be due to the use of different methods used to measure SES and stress [25]. Both SES and stress can be assessed through subjective and objective measures. Subjective measures of socioeconomic status may include opinion surveys that ask participants to rate how much money they or their family has relative to others, whereas objective measures may include assessing adult or parent education levels, occupation, or income. Some studies focus on specific domains of stress such as financial, physical safety, health, or relationship stressors. These dimensions are sometimes measured through subjective measures such as opinion surveys that ask how stressful one’s life is, or through objective measures such as assessing biological markers of cortisol through saliva or hair assays [26].

As previously noted, there are many different sources of stress. Nettle, Andrews, and Bateson [27] postulate that the primary type of stress that motivates excessive food consumption is insecurity due to competition for scarce resources such as food. This theory claims that SES and food insecurity are associated. What creates high insecurity and stress in low SES environments is not the environmental conditions per se, but rather an individual’s low relative standing within the SES hierarchy. Thus, the stressors experienced are often due to the competitive presence of others. Individuals who are of higher status receive preferential treatment and greater access to scarce resources that are needed for survival; those lower down the SES hierarchy would have a chronic sense of insecurity about their current and future ability to acquire scarce resources needed for survival, and therefore have higher chronic stress. What is important to note is that it is not the actual number of resources that are available, but rather the presence of others with higher SES that triggers stress and excessive feeding behaviour.

### 1.2. The Current Study

While previous studies have shown that childhood SES [12] is an important factor that influences the desirability of high- and low-energy-dense food, no study has examined what factors mediate the relationship between childhood SES and food desirability. Therefore, in this study, we investigated the role of stress as an important mediator of this relationship. In addition to stress, we also investigated the roles of current adult SES and trait appetite in this relationship. More specifically, our goal was to address the question, does current adult SES, stress, and trait appetite play a mediating role in the relationship between childhood SES, and the desire for high and low-energy-dense food?

## 2. Methods

### 2.1. Participants

We recruited 311 participants who reside in the United States to participate in this online experiment. Participants were recruited through Qualtrics Panels, an online crowd-sourcing service for human intelligence tasks such as online surveys. Each participant was paid a nominal fee (~$1.00) to participate. Subjects who were younger than 18 years of age, and those who had any form of food allergy, were vegetarian/vegan, or had religious beliefs that influenced their food choices were not permitted to participate in this experiment as these traits would be confounding variables. The participants were 133 men and 178 women, with a mean age of 51.9 years (s.d. = 16.03).

### 2.2. Procedure

Having provided informed consent, the participants were provided with a URL to the online experiment. They were then shown images of 30 food items consisting of five items from each of the major food categories (vegetables, fruits, grains, dairy, meat/poultry, and sweets). The specific food items can be found in Table 1. For further details on the images used in this study, please refer to Swaffield and Gou [14]. With regards to the food items selected to be included in each of the six food categories, there were no inclusion requirements other than to ensure variety. The images were presented in a different randomized order for each participant. Participants viewed each image and rated it according to the following question, which was displayed below the image: “How desirable is this food item to you right now?” Ratings were recorded using a 7-point scale (anchored with the descriptors 1 = extremely undesirable and 7 = extremely desirable).

The caloric value of the 30 food items was determined based on the dataset of Nutritionvalue.org [28]. These food items were split into two equally sized groups based on caloric content per gram. Foods that contained less than 1.49 calories per gram were classified as low-energy-dense foods, and foods greater than 1.50 calories per gram were classified as high-energy-dense foods. For the individual food item’s caloric content per gram, please refer to Table 1.

### 2.3. Measurements

*Trait appetite*. Once the participants rated their preference for each food item, they answered a 15-question General Food Cravings Questionnaire. This questionnaire identified whether the participant had a high or low trait appetite. A person with a high trait appetite is one who has a strong enduring psychological desire for food in the absence of hunger [29,30]. Trait appetite was assessed by the Trait General Food Cravings Questionnaire (α = 0.94), which has been determined to be a reliable and valid measure of general trait-like food cravings [31]. For further details on the construct and discriminant validity of this survey instrument, please refer to [30].

*Past and current socioeconomic status*. Next, the participants were asked 3 questions that would indicate whether they were from a low childhood socioeconomic environment (SES) or a high SES environment. Past childhood SES and current SES were determined by using an established measure [32,33]. Specifically, past childhood SES was determined by asking the participants to indicate how strongly they agreed with the following three statements (α = 0.83): “My family usually had enough money for things when I was growing up”, “I grew up in a relatively wealthy neighbourhood” and, “I felt relatively wealthy compared to the other kids in my school.” The scores from the three questions were averaged to identify whether the participant’s childhood SES was low, medium, or high. Current SES was assessed in a similar manner. The questions asked to determine current SES included (α = 0.83), “I have enough money to buy things I want,” “I don’t need to worry too much about paying my bills,” and “I don’t think I’ll have to worry about money too much in the future.”

*Current level of life stress*. Finally, to determine how stressful the participant’s current life is, they were asked, “how stressful is your life?” Ratings were recorded using a 5-point scale (anchored with the descriptors 1 = very low stress and 5 = very high stress). The rationale for using a single question to assess current stress levels was to avoid an artificial elevation in stress ratings. Specifically, if multiple questions were asked to assess current levels of stress, this could raise the participant’s consciousness of stressful issues that are not top of mind, thus, resulting in feeling their situation is more stressful than it really is.

## 3. Analysis

In order to examine how stress and trait appetite mediates the relationship between childhood SES and food desirability, we conducted a path analysis, which is the conventional statistical method for testing mediation in multiple observed variables. All the analyses were conducted using Mplus 7 [34]. First, descriptive analyses such as means, variances, and correlations among the variables were examined. Then, a path analysis was conducted to examine the relationship between the variables. The model was estimated using robust maximum likelihood in Mplus 7 to correct for any violations of normality in the data set [34]. In assessing the models’ overall fit, we considered the root mean square error of approximation (RMSEA) < 0.08 [35], standardized root mean square residual (SRMR) < 0.08, [36], comparative fit index (CFI) > 0.90 [37], and Chi-Square p-value. The bias-corrected bootstrap confidence interval [38] was used to evaluate the significance of the indirect effect.

## 4. Results

Before we conducted a path analysis, we ran a primary analysis to calculate variable correlations, means, and variance. These results are presented in Table 2.

The conceptual diagram that illustrates the relationship between these variables is shown in Figure 1. The model fit the data very well: Chi-squared = 12.283, df = 9, *p* = 0.20; RMSEA = 0.035; CFI = 0.989; and SRMR = 0.045. In Figure 1, all the coefficients are statistically significant with exception of the link from trait appetite to the desire for low-energy-dense food. This non-significant path is included in Figure 1 because it is related to the hypothesis that predicted that trait appetite would mediate the relationship between childhood SES and the desire for low and high-energy-dense foods. There are several pathways in the model, and we will discuss each of the statistically significant pathways in the following sections.

## 5. The Relationship between Past Childhood SES and the Desire for Low-Energy-Dense Foods

The first pathway (Table 3, path 1a) examined the indirect effect of past (childhood) SES on low-energy-dense food desirability mediated by current adult SES and current stress level.

In path 1a, there is a positive association between childhood SES level and adult desire for low-energy-dense foods. Adult participants who lived in a high SES environment during childhood were more likely to desire low-energy-dense foods than those who lived in a low SES environment during childhood.

Specifically, past (childhood) SES positively and significantly predicted current (adult) SES (0.312 *). Current adult SES then negatively and significantly predicted current stress level (−0.344 *), which negatively and significantly predicted preference for low-energy-dense food (−0.151). Thus, there was an overall positive indirect effect from past SES on the preference for low-energy-dense food. The indirect effect was tested using a bias-corrected bootstrap confidence interval, and the result was significant (95% CI = 0.005, 0.034).

## 6. The Relationship between Past Childhood SES and the Desire for High-Energy-Dense Food

The second analysis examined the indirect effect of past (childhood) SES on the adult desire for high-energy-dense food. This analysis shows there are two paths.

First, path 2a (Table 4) shows that when the mediating effect of trait appetite (desire for food in the absence of an energy deficit) is factored out from the analysis, adult participants who had high childhood SES were more likely to have a high preference for high-energy-dense food.

Specifically, in path 2a, past childhood SES positively and significantly predicted the desire for high-energy-dense food whereas the second path (Table 4, path 2b), shows an opposite effect whereby past (childhood) SES had a significant negative effect on the desire for high-energy-dense food, meaning when trait appetite is factored into the analysis, if an adult study participant had high childhood SES, they were more likely to have low trait appetite and a low desire for high-energy-dense food. In contrast, if an adult participant had low childhood SES, they were more likely to have a high trait appetite and a high desire for high-energy-dense food.

The specific details are as follows. In path 2a, past (childhood) SES positively and significantly predicted current (adult) SES (0.312). Current adult SES then negatively and significantly predicted current stress level (−0.344), which negatively and significantly predicted the desire for high-energy-dense food (−0.102). Thus, there was an overall positive indirect effect from past SES to preference for high-energy-dense food. The indirect effect was tested using a bias-corrected bootstrap confidence interval, and the result was significant (95% CI = 0.002, 0.027). Thus, if a participant had high childhood SES, adult participants were more likely to have a high preference for high-energy-dense food.

The second pathway (2b) included trait appetite as a mediating variable. The overall indirect effect of past (childhood) SES on the desire for high-energy-dense food when mediated by adult SES, stress level, and trait appetite was negative. The indirect effect was tested using a bias-corrected bootstrap confidence interval, and the result was significant (95% CI = −0.014, −0.003).

The path model presented in Figure 1 explained 27.2% of the variance in stress, 3.8% of the variance in the preference for low-energy-dense food, 11.2% of the variance in the preference for high-energy-dense food, 9.6% of the variance in trait appetite, and 11.1% of the current SES. The results showed that the path model is more efficient at predicting participants’ preferences for high-energy-dense food than participants’ preferences for low-energy-dense food.

In sum, after combining the positive and negative effects, stress has an overall significant positive effect on the desire for high-energy-dense food, with a value of 0.081 (bootstrap CI: 0.034, 0.129). However, stress does not have a significant effect on the desire for low-energy-dense food, with a value of 0.028 (bootstrap CI: −0.10, 0.066).

## 7. Discussion

It is often believed that healthy eating is about making wise choices regarding what and how much is eaten. This perspective assumes eating behaviour is a cognitive choice. However, eating behaviour is deeply rooted within our evolved biology and is emotionally driven. Ramos and Olden [39] state that this concept is sometimes explained by the metaphor, “genetics loads the gun, but the environment pulls the trigger.” To understand the phenomenon of food consumption, we need to understand how evolved biological traits such as tastebuds, dopamine production, cortisol, and other hormones interact with environmental factors such as stress.

This study was designed to examine the environmental trigger of stress and its impact on food consumption. What is more, we examined whether stress and trait appetite mediate the relationship between childhood SES and adult desire for low- and high-energy-dense foods. Through path analysis, it was discovered that both stress and trait appetite are indeed mediators. Furthermore, we found that trait appetite also mediates the relationship between stress and the desire for high-energy-dense foods. Specifically, elevated stress only increases the desire for high-energy-dense foods through trait appetite; whereas, if trait appetite is kept constant, stress has a negative impact on the desire for high-energy-dense food.

There are three dominant theories as to why stress triggers the desire to eat. The first theory is the insurance hypothesis. The insurance hypothesis postulates that humans have an instinctive drive to overconsume food to acquire excess fat stores to buffer the impact of future famine. It is believed that as environmental conditions become harsh, a sense of resource scarcity is felt, which creates a sense of stress. To reduce stress, humans are driven to acquire and consume excess food [27].

The second reason stress can increase the desire for high-energy-dense foods is because stress signals to the body that it needs to prepare for a fight or flight response. As Sapolsky [40] states, stress is an outcome of the body being knocked out of a state of homeostasis. When this happens, stress triggers a number of physiological responses that produce a drive to restore the body to a homeostatic state. At the most fundamental level, the purpose of the stress response is to prepare the body for a fight or flight response. To fuel this response, our muscles need to mobilize stored glucose and fats as quickly as possible [40].

The final explanation as to why stress increases the desire for high-energy-dense foods and less so for low-energy-dense foods is because energy-dense foods trigger the production of the neurotransmitter dopamine. Dopamine production can blunt the stress response, which promotes the return to a homeostatic state. This, in turn, makes the stressed individual feel better [24]. The desire for energy-dense foods is also reinforced through the evolution of human tastebuds. Humans have tastebuds for sweet, umami (fat), salty, bitter, and sour foods. Humans do not have tastebuds for low-energy-dense foods, which are experienced as tasting bland. Bland-tasting foods do not trigger dopamine production without adding energy-dense supplements such as butter and sauces and dressings [41].

While these theories have merit, they do not fully explain why, under stress, some people experience an increase in appetite whereas others experience a decrease. One possible reason is that different types of environmental stressors have different effects on appetite. As noted by [14], financial stress tends to increase appetite whereas concerns about physical safety tend to decrease appetite.

The second reason stress may increase or decrease the desire for high-energy-dense foods may be due to some environments having different stressor intensities. As previously mentioned, low-level chronic stress tends to increase appetite [15,16], whereas acute high-intensity stress tends to decrease appetite [16,17,18,19].

Three limitations of this study should be noted. First, at the time the study was designed, we considered the question, will the participant’s current stated level of stress be a product of enduring chronic stress or is it momentary stress triggered by an event such as receiving bad news? Another consideration was, if multiple questions are asked that raise the participant’s consciousness of stressful issues that are not currently top of mind, could this artificially elevate the participant’s stated level of stress? Thus, it would be beneficial to ensure that the method used to measure stress is indeed measuring chronic stress rather than momentary stress. Second, different types of events trigger different stress responses. It may be wise to include questions that help to identify what types of stressors the participants are feeling. Third, it is well-established that how one responds to stress is mediated and moderated by the intensity of the stressor. Therefore, it would be beneficial to control for the intensity of the stressors so that it can be determined whether the stressor is perceived to be chronic (low intensity) or acute (high intensity).

## 8. Conclusions

In conclusion, this study shows the complexity of the relationship between psychosocial variables and eating behaviour. It also shows that, by themselves, SES level and stress are insufficient in explaining food consumption patterns. Being stressed is both a psychological and physiological state that can be triggered by different environmental conditions. Furthermore, how an adult responds to environmental stressors is influenced by the socioeconomic conditions that were experienced during childhood.

From a practical standpoint, the findings from this study have implications for those involved in health education and nutrition counseling. In contrast to the perspective that healthy eating is simply a product of proper education, practitioners in these fields would be wise to acknowledge that the motivation to eat and the selection of energy-dense foods is a product of an interaction between one’s evolved biology and environmental stressors. It may also be beneficial for educators and health practitioners to incorporate stress management strategies into programs aimed at developing healthy eating habits.

## Figures and Tables

**Figure 1 behavsci-12-00202-f001:**
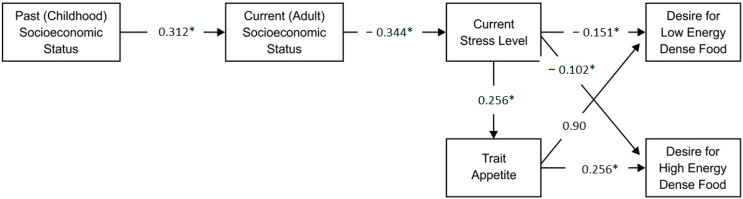
Path analysis for low and high-energy-dense food preferences. * Denotes path is significant. Footnote: “Path coefficients” are partial regression coefficients. Partial regression coefficients remove the effects of correlated influences (or the effect of two independent variables on a dependent variable). For example, stress (IV) and low SES (IV) are all correlated with obesity (DV). Each of the IVs are called ‘partial’ coefficients because they remove the effect of the additional IV variables.

**Table 1 behavsci-12-00202-t001:** Standard energy content of the presented food items (in calories per gram).

Low Energy	Cal/g	High Energy	Cal/g
Foods		Foods	
Celery	0.14	Spaghetti	1.57
Carrots	0.41	Ham	1.88
Pear	0.42	Chicken	2.18
Orange	0.47	Steak	2.50
Milk	0.50	Bread	2.70
Apple	0.52	Hamburger	2.88
Yogurt	0.56	Pastry	3.40
Blueberries	0.57	Candy	3.60
Oatmeal	0.58	Cheese	4.10
Potato	0.71	Cupcake	4.25
Peas	0.81	Bacon	4.62
Cottage cheese	0.84	Chocolate	5.10
Corn	0.86	Cookies	5.40
Banana	0.89	Nuts	5.77
Rice	1.11	Butter	7.17

**Table 2 behavsci-12-00202-t002:** Correlations among model constructs.

Variable	1	2	3	4	5	6
1. Stress level	-					
2. Desire for low-energy-dense foods	−0.158 **	-				
3. Desire for high-energy-dense foods	−0.046	0.661 **	-			
4. Trait appetite	0.238 **	0.074	0.311 **	-		
5. Past childhood SES	−0.124 *	0.170 **	0.157 **	0.086	-	
6. Current childhood SES	−0.459 **	0.083	0.041	−0.079	0.268 **	-
Mean	2.761	4.371	4.841	2.88	3.53	4.003
Variance	1.476	0.98	0.958	1.697	1.945	2.635

* Significant at 0.05, ** Significant at 0.01.

**Table 3 behavsci-12-00202-t003:** The indirect effect of past (childhood) SES on low-energy-dense food desirability.

								Bootstrap CI
1a.	Past SES	→	Current SES	→	Current Stress Level	→	Desire for LEDF	0.005, 0.034

LEDF = low energy dense foods.

**Table 4 behavsci-12-00202-t004:** The indirect effect of past (childhood) SES on the adult desire for high-energy-dense food.

										Bootstrap CI
2a.	Past SES	→	Current SES	→	Current Stress Level	→			Desire for HEDF	0.002, 0.027
2b.	Past SES	→	Current SES	→	Current Stress Level	→	Trait Appetite	→	Desire for HEDF	− 0.014, − 0.003

HEDF = high energy dense foods.

## Data Availability

The datasets generated during and/or analyzed during the current study are available from the corresponding author on reasonable request.

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
