# Peer review of "How Childhood Socioeconomic Status Impacts Adult Food Preference: The Mediating Role of Stress and Trait Appetite"

_behavsci, 2022, doi:10.3390/bs12070202_

Round 1
Reviewer 1 Report
Dear authors, thanks for sending your contribution to the Journal of Behavioral Sciences. Your work is related to one a specific field in pediatric nutrition and eating behavior. But there are some comments that you must address to improve your work.
1. Methods. Please add in this section the registration number of the ethics committee which approved your work.
2. Please explain why you didn’t use a validated questionnaire to measure the current level of life stress? Are you sure about the correct selection method to measure this variable?
3. Results. Can you add a table with the description data of your sample? Because we only know the number of participants, age, and the proportion of sex. Maybe there is other data that could affect eating behavior. Also, could be very interesting if you can re-analyze your data and add some other variables for example sex, education, diet history, and diseases that could affect the eating behavior.
4. Discussion. Please add more discussion about the importance of our results in the current literature, also explain your results more specifically, not only with the theories around.
5. The reference style is not correct to the author's instructions. Please resolve it.
Author Response
Reviewer 1
Please note, the editor has requested that the edits be made and resubmitted within a 10-day period. Working within this short time frame, we submit the follow edited paper and comments. We graciously that you for your effort in helping us improve the quality of this paper.
Also, please note, there are very minor changes to 3 statistics in the path analysis model as shown in Figure 1. These edits did not change the any of the conclusions. They simply increased the precision of the model.
- Methods. Please add in this section the registration number of the ethics committee which approved your work.
We have added the ethics approval reference number from the University of Alberta which is Pro00043569_REN1.
I have a letter of ethics approval in the form of an e-mail from the University of Stirling, but there is no reference number. This document has already been provided to the assistant editor Miss Ioana Serban.
- Please explain why you didn’t use a validated questionnaire to measure the current level of life stress? Are you sure about the correct selection method to measure this variable?
At the time of the study design, we did talk about how to measure current stress levels. What we wanted to do was capture the participants feeling of stress at the very moment that they were participating in the study. We are conscious of the fact that if we asked too many stress-related questions, the very act of thinking of a list of issues could artificially increase the participant’s stress rating.
Second, the data obtained showed that the question asked was sensitive enough to capture variations in perceptions of current stress levels. Therefore, we feel comfortable with the method that we used.
We have added a note in the Limitations section of the paper. See lines 304-316.
Specifically,
Three limitations of this study should be noted. First, at the time the study was being designed, we considered the question, will the participant’s current stated level of stress be a product of enduring chronic stress or, is it momentary stress triggered by an event such as receiving bad news? Another consideration was, if multiple questions are asked that raise the participant’s consciousness of stressful issues that are not currently top of mind, could this artificially elevate the participant’s stated level of stress? Thus, it would be beneficial to ensure that the method used to measure stress is measuring chronic stress rather than momentary stress.
Second, different types of events trigger different stress responses. It may be wise to include questions that help to identify what types of stressors the participants are feeling. Third, it is well established that how one responds to stress is mediated and moderated by the intensity of the stressor. Therefore, it would be beneficial to control for the intensity of the stressors so that it can be determined whether the stressor is perceived to be chronic (low intensity) or acute (high intensity). “
Also, stress levels can be assessed through biomarkers such as cortisol samples as well as through verbal statements. Unfortunately, adult cortisol samples cannot be used to measure childhood stress levels that were experienced many years earlier. The best we can do is infer a participant’s early childhood stress levels based on their reflections of what their early childhood conditions were like.
The three questions that were asked to infer the level of early childhood stress include:
- My family usually had enough money for things when I was growing up.
- I grew up in a relatively wealthy neighborhood.
- I felt relatively wealthy compared to other kids in my school.
Two concepts are particularly relevant here. First is the concept of chronic financial stress and second, the concept of relativity to others who have more.
Question 1 assesses whether the participant’s homelife was overshadowed with stress due to chronic financial challenges. The relevance of this question is supported in the paper by Swaffield and Guo (2020). In this paper, the researchers show experimentally how financial stress increases appetite. Here is the reference for this paper.
Swaffield, J. B., & Guo, Q. (2020). Environmental stress effects on appetite: Changing desire for high- and low-energy foods depends on the nature of the perceived threat. Evolution, Mind and Behaviour. doi:10.1556/2050.2018.00008.
The second concept of “relativity to others” is also an important factor in assessing early childhood stress and is assessed via questions 2 and 3. Stress researcher Robert Sapolsky (2018) reinforces this point in the statement,
“low socioeconomic status does not necessarily lead to elevated stress, but rather, low status generates elevated levels of stress, when in the presence of others who have higher status.”
Sapolsky, R. M. (2018). Behave: The biology of humans at our best and worst. NY, NY: Penguin Books.
There is also a strong theoretical basis that explains why relativity is an important factor when assessing stress. Briefly, it has to do with the fact that life is competitive. We compete for scarce resources, jobs, mates and so on. Suffice it to say, those who have relatively less, have the highest stress levels and highest cortisol levels.
These concepts have been noted in the sentences identified by line number 104 to 113.
- Results. Can you add a table with the description data of your sample? Because we only know the number of participants, age, and the proportion of sex. Maybe there is other data that could affect eating behavior. Also, could be very interesting if you can reanalyze your data and add some other variables for example sex, education, diet history, and diseases that could affect the eating behavior.
The only other data that we collected was information on whether the participant was a vegan or vegetarian, had food allergies or religious beliefs that could affect their food choices. If a participant said that one of these variables did apply to them, they were removed from the study because these traits are considered confounding variables.
Unfortunately, no other information was collected regarding diet history, education or diseases. We did however collect information on sex and this identified on the paper.
- Please add more discussion about the importance of our results in the current literature, also explain your results more specifically, not only with the theories around.
The following write-ups have been added to the discussion section.
(Lines 279-286)
It often believed that healthy eating is about making wise choices regarding what and how much is eaten. This perspective assumes eating behavior is a cognitive choice. However, eating behavior is deeply rooted within our evolved biology and is emotionally driven. Ramos and Olden (2008) state, that this concept is sometimes explained by the metaphor, “genetics loads the gun, but the environment pulls the trigger.” To understand the phenomenon of food consumption, we need to understand the how evolved biological traits such as such as tastebuds, dopamine production, cortisol and other hormones interact with environmental factors such as stress.
(Lines 310-318)
A final explanation as to why stress increases the desire for energy dense foods and less so for low energy dense foods, is because energy dense foods trigger the production of the neurotransmitter dopamine. Dopamine production can blunt the stress response which promotes a return to a homeostatic state. This in turn, makes the stressed individual feel better (Sapolsky, 2018). The desire for energy dense foods is also reinforced through the evolution of human tastebuds. Humans have taste buds for sweet, umami (fat), salty, bitter and sour foods. Humans do not have tastebuds for low energy dense foods that are experienced as tasting bland. Bland tasting foods do not trigger dopamine production without adding supplements such as butter and sauces and dressings (Breslin, 2013).
Add part about understanding that the desire to eat is not simply a matter of making wise choices regarding what to eat and how much.
We have also added an additional paragraph to the conclusion. Please see lines 351-356.
- The reference style is not correct to the author's instructions. Please resolve it.
The referencing issues have been corrected.
Reviewer 2 Report
This article presents the relationship of childhood socioeconomic status impacts adult food preference.
Thank you for possibility to review the manuscript on very interesting and important topic – the mediating role of stress and trait appetite in relationship of SES and the desire for high and low energy dense food and obesity.
Abstract – it should be presented how hypothesis was supported by statistical results.
Methods –
This sentence is related to results and should be moved to part Results: The participants were 133 men and 178 women, with a mean age of 51.9 years (sd±16.03). (Line 132-133)
What is the reference for method used for measurement of Current level of life stress? It is very difficult to reliably assess stress level with just one subjective question.
Accurate applying of bootstrapping correction
Results –
Please, clarify variable Current childhood SES
Discussion – line 285 and 289 are very similar.
The discussion should be deepened because there are probably other factors that are relevant to the relationship between SES and desire for high energy foods. For example, was the timing of questionnaire completion - post-feeding or fasting - taken into account? It is possible that the same person may give different answers depending on the momentary state of completing the questionnaire.
Author Response
Please note, the editor has requested that the edits be made and resubmitted within a 10-day period. Working within this short time frame, we submit the follow edited paper and comments. We graciously that you for your effort in helping us improve the quality of this paper.
Also, please note, there are very minor changes to 3 statistics in the path analysis model as shown in Figure 1. These edits did not change the any of the conclusions. They simply increased the precision of the model.
- Abstract – it should be presented how hypothesis was supported by statistical results.
We have mentioned in the abstract that the results were statistically significant.
- Methods – This sentence is related to results and should be moved to part Results: The participants were 133 men and 178 women, with a mean age of 51.9 years (sd±16.03). (Line 132-133) What is the reference for method used for measurement of Current level of life stress? It is very difficult to reliably assess stress level with just one subjective question. Accurate applying of bootstrapping correction
In a typical path analysis study the data mentioned is normally included in the methods section rather than the results section. This is also consistent with other path analysis articles in the journal Behavioral Sciences. For example, the article Buffering the Fear of COVID-19: Social Connectedness Mediates the Relationship between Fear of COVID-19 and Psychological Wellbeing, (Humphrey et al., March 2022) also included this data in the methods section.
With regards to the question, “What is the reference for method used for measurement of Current level of life stress? It is very difficult to reliably assess stress level with just one subjective question.” Please see the write-up shown in the next section 3. Results. Thank you.
- Results –
3.1 Please, clarify variable Current childhood SES
We are not clear on what is being asked in this question as there is only “past childhood socioeconomic status” and “current adult status.” However, we will try to provide an explanation here that we hope will answer your question.
The best we can do is infer a participant’s early childhood stress levels based on their reflections of what their early childhood conditions were like.
The three questions that were asked to infer the level of early childhood stress include:
- My family usually had enough money for things when I was growing up.
- I grew up in a relatively wealthy neighborhood.
- I felt relatively wealthy compared to other kids in my school.
Two concepts are particularly relevant here. First is the concept of chronic financial stress and second, the concept of relativity to others who have more.
Question 1 assesses whether the participant’s homelife was overshadowed with stress due to chronic financial challenges. The relevance of this question is supported in the paper by Swaffield and Guo (2020). In this paper, the researchers show experimentally how financial stress increases appetite. Here is the reference for this paper.
Swaffield, J. B., & Guo, Q. (2020). Environmental stress effects on appetite: Changing desire for high- and low-energy foods depends on the nature of the perceived threat. Evolution, Mind and Behaviour. doi:10.1556/2050.2018.00008.
The second concept of “relativity to others” is also an important factor in assessing early childhood stress and is assessed via questions 2 and 3. Stress researcher Robert Sapolsky (2018) reinforces this point in the statement,
“low socioeconomic status does not necessarily lead to elevated stress, but rather, low status generates elevated levels of stress, when in the presence of others who have higher status.”
Sapolsky, R. M. (2018). Behave: The biology of humans at our best and worst. NY, NY: Penguin Books.
There is also a strong theoretical basis that explains why relativity is an important factor when assessing stress. Briefly, it has to do with the fact that life is competitive. We compete for scarce resources, jobs, mates and so on. Suffice it to say, those who have relatively less, have the highest stress levels and highest cortisol levels.
These concepts have been noted in the sentences identified by line number 104 to 113.
- Discussion –
4.1 line 285 and 289 are very similar.
The repeated statement regarding preparing the body for a fight or flight response has been edited/removed.
4.2 The discussion should be deepened because there are probably other factors that are relevant to the relationship between SES and desire for high energy foods. For example, was the timing of questionnaire completion - post-feeding or fasting - taken into account? It is possible that the same person may give different answers depending on the momentary state of completing the questionnaire.
Regarding the timing of the questionnaire being answered. You are correct, the factors mentioned could influence how a participant responds to the questions in the experiment. We do however think that the following factors would have minimized the issues mentioned.
First, the experiment was administered online, and the participants could participate at any time of the day or night. Some people participated in the morning, some participate in the afternoon, and some participated in the evening.
Second, the sample size was 311 people. This large sample should minimize the effects of the confounding variables mentioned.
We have also added to the discussion the following paragraphs:
(Lines 279-286)
It often believed that healthy eating is about making wise choices regarding what and how much is eaten. This perspective assumes eating behavior is a cognitive choice. However, eating behavior is deeply rooted within our evolved biology and is emotionally driven. Ramos and Olden (2008) state, that this concept is sometimes explained by the metaphor: “genetics loads the gun, but the environment pulls the trigger.” To understand the phenomenon of food consumption, we need to understand the how evolved biological traits such as such as tastebuds, dopamine production, cortisol and other hormones interact with environmental factors such as stress.
(Lines 310-318)
A final explanation as to why stress increases the desire for energy dense foods and less so for low energy dense foods, is because energy dense foods trigger the production of the neurotransmitter dopamine. Dopamine production can blunt the stress response which promotes a return to a homeostatic state. This in turn, makes the stressed individual feel better (Sapolsky, 2018). The desire for energy dense foods is also reinforced through the evolution of human tastebuds. Humans have taste buds for sweet, umami (fat), salty, bitter and sour foods. Humans do not have tastebuds for low energy dense foods that are experienced as tasting bland. Bland tasting foods do not trigger dopamine production without adding supplements such as butter and sauces and dressings (Breslin, 2013).
We have also added an additional paragraph to the conclusion.
Reviewer 3 Report
The age of participants is relatively high being 51 with a stdev of 18. This means the youngest would be 33. Is there a reason why the age group is this high?
Please include ethics statement for this study
Instead of referring to a reference for a study protocol — I’d like to invite the author to summarise it in a few sentences for the readers
Is there a reason if the selection of food items on table 1?
Stress scale - where is it from? Stress measurement can be complex and may not only be derived from one scaling item.
One would expect a inverse relationship but this wasn’t the case in the level and trait appetite where a weak and strong relationship was observed. Any explanation why this might’ve happened?
Discussion needs to be rewritten accordingly to the key results that the authors found - currently it feels very generalised.
Author Response
Please note, the editor has requested that the edits be made and resubmitted within a 10-day period. Working within this short time frame, we submit the follow edited paper and comments. We graciously that you for your effort in helping us improve the quality of this paper.
Also, please note, there are very minor changes to 3 statistics in the path analysis model as shown in Figure 1. These edits did not change the any of the conclusions. They simply increased the precision of the model.
The age of participants is relatively high being 51 with a stdev of 18. This means the youngest would be 33. Is there a reason why the age group is this high?
Response: At the time participants were recruited for this study, the only limitation on age was that they had to be 18 years old or older. The participants were recruited through the Qualtrics online crowd sourcing platform. Qualtrics sent an e-mail out to thousands of people who were registered panel members that were at least 18 years old. Thus, the age of participants is a product of those who responded to the request for participation.
- Please include ethics statement for this study instead of referring to a reference for a study protocol.
We are not sure what is meant by “study protocol.” In North America the term study protocol refers to a research proposal summary. However, we looked at the author guidelines and wonder if perhaps you are referring to including a statement regarding including a Declaration of Helsinki statement? If yes, we have included this statement. See below:
Ethics approval. All research methods were conducted in accordance with the Declaration of Helsinki and the protocol was approved by the Ethics Review Board at University of Stirling via a letter of approval, and by the University of Alberta Ethics Review Board, registration number Pro00043569_REN1.
Informed consent. All participants formally gave consent to participate in this study. In addition, participants were required to confirm that they were agreeing to submit their results so that they could be included in this study.
We also looked at other papers in the same journal and I believe that we are consistent with similar papers that have used a path analysis. That being said, if our assumption of what is being asked is still incorrect, please let us know and we will do our best to make the appropriate edits. Please note, the editor has requested that we return the paper with edits in only 10 days, so this would have to be done quickly to meet editor expectations.
Instead of referring to a reference for a study protocol – I’d like to invite the author to summarise it in a sentence for the readers.
We believe this question is related to the second question. Thus, if you would like us to do something different than the edits made to the Ethics Approval and Informed consent sections, please advise and we will do what we can to make the appropriate edits.
I’d like to invite the author to summarise it in a few sentences for the readers Is there a reason if the selection of food items on table 1?
Response: Other than selecting 5 foods from each of the 6 categories (fruits, vegetables, meats/poultry, grains, dairy, and sweets), there was no rationale for the food items that were included within each category.
We added the statement, “With regards to the food items selected to be included in each of the six food categories, there was no inclusion requirements other than to ensure variety. (see lines 140-143)
- Stress scale - where is it from? Stress measurement can be complex and may not only be derived from one scaling item.
Your point is well taken. At the time of the study design, we did talk about how to measure current stress levels. What we wanted to do was capture the participants feeling of stress at the very moment that they were participating in the study. We are conscious of the fact that if we asked too many stress-related questions, the very act of thinking of a list of issues could artificially increase the participant’s stress rating.
Second, the data obtained showed that the question asked was sensitive enough to capture variations in perceptions of current stress levels. Therefore, we feel comfortable with the method that we used.
We have added a note in the Limitations section of the paper. See lines 304-316.
Specifically,
Three limitations of this study should be noted. First, at the time the study was being designed, we considered the question, will the participant’s current stated level of stress be a product of enduring chronic stress or, is it momentary stress triggered by an event such as receiving bad news? Another consideration was, if multiple questions are asked that raise the participant’s consciousness of stressful issues that are not currently top of mind, could this artificially elevate the participant’s stated level of stress? Thus, it would be beneficial to ensure that the method used to measure stress is measuring chronic stress rather than momentary stress.
Second, different types of events trigger different stress responses. It may be wise to include questions that help to identify what types of stressors the participants are feeling. Third, it is well established that how one responds to stress is mediated and moderated by the intensity of the stressor. Therefore, it would be beneficial to control for the intensity of the stressors so that it can be determined whether the stressor is perceived to be chronic (low intensity) or acute (high intensity). “
Also, stress levels can be assessed through biomarkers such as cortisol samples as well as through verbal statements. Unfortunately, adult cortisol samples cannot be used to measure childhood stress levels that were experienced many years earlier. The best we can do is infer a participant’s early childhood stress levels based on their reflections of what their early childhood conditions were like.
The three questions that were asked to infer the level of early childhood stress include:
- My family usually had enough money for things when I was growing up.
- I grew up in a relatively wealthy neighborhood.
- I felt relatively wealthy compared to other kids in my school.
Two concepts are particularly relevant here. First is the concept of chronic financial stress and second, the concept of relativity to others who have more.
Question 1 assesses whether the participant’s homelife was overshadowed with stress due to chronic financial challenges. The relevance of this question is supported in the paper by Swaffield and Guo (2020). In this paper, the researchers show experimentally how financial stress increases appetite. Here is the reference for this paper.
Swaffield, J. B., & Guo, Q. (2020). Environmental stress effects on appetite: Changing desire for high- and low-energy foods depends on the nature of the perceived threat. Evolution, Mind and Behaviour. doi:10.1556/2050.2018.00008.
The second concept of “relativity to others” is also an important factor in assessing early childhood stress and is assessed via questions 2 and 3. Stress researcher Robert Sapolsky (2018) reinforces this point in the statement,
“low socioeconomic status does not necessarily lead to elevated stress, but rather, low status generates elevated levels of stress, when in the presence of others who have higher status.”
Sapolsky, R. M. (2018). Behave: The biology of humans at our best and worst. NY, NY: Penguin Books.
There is also a strong theoretical basis that explains why relativity is an important factor when assessing stress. Briefly, it has to do with the fact that life is competitive. We compete for scarce resources, jobs, mates and so on. Suffice it to say, those who have relatively less, have the highest stress levels and highest cortisol levels.
These concepts have been noted in the sentences identified by line number 104 to 113.
- One would expect a inverse relationship but this wasn’t the case in the level and trait appetite where a weak and strong relationship was observed. Any explanation why this might’ve happened?
The second paragraph as identified by lines 310-318 in the article answer the question that you have asked. In addition, we have added lines 279-286.
(Lines 279-286)
It often believed that healthy eating is about making wise choices regarding what and how much is eaten. This perspective assumes eating behavior is a cognitive choice. However, eating behavior is deeply rooted within our evolved biology and is emotionally driven. Ramos and Olden (2008) state, that this concept is sometimes explained by the metaphor: “genetics loads the gun, but the environment pulls the trigger.” To understand the phenomenon of food consumption, we need to understand the how evolved biological traits such as such as tastebuds, dopamine production, cortisol and other hormones interact with environmental factors such as stress.
(Lines 310-318)
A final explanation as to why stress increases the desire for energy dense foods and less so for low energy dense foods, is because energy dense foods trigger the production of the neurotransmitter dopamine. Dopamine production can blunt the stress response which promotes a return to a homeostatic state. This in turn, makes the stressed individual feel better (Sapolsky, 2018). The desire for energy dense foods is also reinforced through the evolution of human tastebuds. Humans have tastebuds for sweet, umami (fat), salty, bitter and sour foods. Humans do not have tastebuds for low energy dense foods that are experienced as tasting bland. Bland tasting foods do not trigger dopamine production without adding energy dense supplements such as butter and sauces and dressings (Breslin, 2013).
The answer to this question also relates to your point made in #6 below.
- Discussion needs to be rewritten accordingly to the key results that the authors found - currently it feels very generalised.
We have added the section as noted above.
Also, we have also added an additional paragraph to the conclusion.
Round 2
Reviewer 1 Report
Dear authors, thanks for answer our comments.
Author Response
Thank you very much for your feedback and and support. As no specific edits have been requested, the manuscript has not been edited.
Best regards, JS
Reviewer 3 Report
On study protocol, its not about the ethics statement - but rather the methodologies that was applied in this study.
It seems that the selection of food items were just selected from the authors POV. Have the authors considered the familiarity or neophobia in some food items or is it just common food items?
Fair statement re stress level, the authors seems to think about this thoroughly
Author Response
Dear Reviewer, thank you for your time, thoughts and feedback on our manuscript. We awknowledge that there are always improvements that can be made - unfortunately, we often discover many of these ideas for improvement after the research study has been implemented and hard work has been completed. We also struggle to find that happy medium in knowing how much detail to include. All that being said, we are confident that this paper is theoretically and methodologically robust. The principle investigator of this study has been researching environmental harsheness and food consumption since 2011, so while there will always be spots for improvements, we think that we have covered the "must have" and many of the "nice to have" concepts in the paper. We hope that you agree.
With regards to the comment of including a protocol. We did review the journal's guidelines and I believe that the methods section is comprehensive and should meet the protocol standards.
With regards to the selection of the food items that were included inthis paper, we tried to include foods that people would be familar with and would have access to.
Lastly, with regards to neophobia, we did not look at this phenomenon.
With humans neophobia can apply to infants, children, adolescents and adults. However, most of the research in this area is with infants and children. I think the foods that we used in this study are all pretty mainstream foods (apples, oranges, carrots, etc.), so I am not too concerned that the foods would be novel or unfamilar to most people. In the even that there was a few participants who were unfamilar with any of the foods, the very large sample size (311 participants) should minimize the effect of a participant having food neophobia.
Once again, we sincerely than you for your assistance in fine-tuning and improving our paper. Sincerely, JS
PS: As there were no specific edits requested, the paper has not been editied.